# Many-body Bell inequalities for bosonic qubits

**Jan Chwedeńczuk**

Faculty of Physics, University of Warsaw, ul. Pasteura 5, PL–02–093 Warszawa, Poland

## Abstract

Since John Bell formulated his paramount inequality for a pair of spin-1/2 particles, quantum mechanics has been confronted with the postulates of local realism with various equivalent configurations. Current technology, with its advanced manipulation and detection methods, allows to extend the Bell tests to more complex structures. The aim of this work is to analyze a set of Bell inequalities suitable for a possibly broad family of many-body systems with the focus on bosonic qubits. We develop a method that allows for a step-by-step study of the many-body Bell correlations, for instance among atoms forming a two-mode Bose-Einstein condensate or between photons obtained from the parametric-down conversion. The presented approach is valid both for cases of fixed and non-fixed number of particles, hence it allows for a thorough analysis of quantum correlations in a variety of many-body systems.

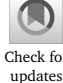

# 1 Introduction

The consequences of the quantum-mechanical description of the physical world are often puzzling or even troubling. In their famous paper dating back to 1935 [1], Albert Einstein, Boris Podolsky and Nathan Rosen (EPR) investigated one such counter-intuitive prediction, related to what Einstein called the "spooky action at the distance". The EPR trio asked if the theory that allows for such bizarre phenomena should be extended to a more complete model of reality. Later that year, Erwin Schrödinger introduced the concept of entanglement, as a fundamental ingredient of those exotic and apparently non-local effects [2]. Only 29 years later John Bell showed that there are quantum states, such as a singlet state of two spin-1/2 particles, that contain correlations too strong for a locally realistic theory. Hence, the extension of quantum mechanics, suggested by the EPR trio, to a theory that would be consistent with the postulates of local realism, is not possible [3].

The Bell's achievement is important because of its implications for our understanding of quantum mechanics, but also because it provides a rather simple recipe for testing the non-locality of the theory. Its experimentally-friendly version [4], known as the CHSH inequality (John Clauser, Michael Horne, Abner Shimony and Richard Holt) has been used in numerous configurations, some prominent examples are [5–16] while [17] contains more references. Recently, Bell tests closing the loopholes, including that related to the spatio-temporal separation, have been reported [18–21].

These experiments operate on pairs of entangled qubits: photons obtained in the parametric-down conversion [22–24], electrons, Josephson junctions [25] or nitrogen vacancies [26]. Now that we have gained extensive control over complex quantum systems, Bell correlations can be tested on the many-body level. This calls for a new set of Bell inequalities, tailored for such purposes, and a versatile method has been developed in [27–29]. It assumes that the system is composed of a collection of qubits and allows to investigate the many-body entanglement, EPR-steering and Bell correlations in a systematic way [30, 31].

Here we adapt these inequalities to bosonic systems and provide a tool for detecting the Bell correlations with correlation functions of any desired order. The presented method could be implemented in Bose-Einstein condensates (BECs) trapped in a double-well potential or in an optical lattice [32]. We show how to extend this approach to systems where the number of particles is not fixed. This way, it could be used to detect the entanglement and the Bell correlations in a multi-photon parametric-down-conversion state, without the need to restrict to a single pair of photons. Similarly, an ideal test ground for these inequalities would be a collection of atomic pairs, scattered from a BEC [33–40].

This manuscript is organized as follows. We start with Section 2, where we introduce the theoretical tools necessary to analyze many-body Bell correlations in systems of bosonic qubits. We introduce a family of many-body Bell inequalities in Secction 2.1. First, we focus on the case, when parties can be addressed individually, see Section 2.1.1. Next, in Section 2.1.2 we consider the permutationally invariant case. In Section 2.1.3 we discuss the case when the parties can be labeled with additional degree(s) of freedom. In Section 2.2 we show how these inequalities can be tested with quantum systems. In particular, in Section 2.2.3 we show how to adapt these tools to the case when the number of qubits is not fixed. Next, we move to Section 3, where we apply these techniques to some prominent examples: a BEC confined in a double-well potential (see Section 3.1) or a multi-photon parametric-down conversion signal (Section 3.2). We present some closing remarks in Section 4 and conclude with Section 5.

## 2 Theoretical toolbox

In this section we develop all the theoretical tools necessary to systematically analyze Bell correlations in complex many-body systems of bosonic qubits.

### 2.1 Many-body Bell inequalities

We begin with a discussion of a broad family of Bell inequalities that can be used in systems of individually accessible qubits, such as spin chains. Later, we generalize these inequalities to systems of identical particles that cannot be addressed one-by-one.

#### 2.1.1 Individually addressable parties

Consider a pair of binary physical quantities: $\sigma_1^{(k)} = \pm 1$ and $\sigma_2^{(k)} = \pm 1$, independently measured for $m$ particles (thus $k \in \{1, \ldots . m\}$). Take the product of these outcomes

$$\Sigma_m = \prod_{k=1}^{m} \sigma_{s_k}^{(k)}, \tag{1}$$

where $\sigma_{s_k}^{(k)} = \frac{1}{2}(\sigma_1^{(k)} + (s_k)i\sigma_2^{(k)})$ and the sign $s_k = \pm 1$ is chosen by each party separately. If the average of $\Sigma_m$ over many experimental realizations is consistent with a local realistic theory, it can be reproduced by taking a mean over some probability distribution $p(\lambda)$ of a "hidden variable" $\lambda$, namely

$$\langle \Sigma_m \rangle = \int d\lambda\, p(\lambda) \prod_{k=1}^{m} \sigma_{s_k}^{(k)}(\lambda). \tag{2}$$

As $|\sigma_{s_k}^{(k)}|^2 = \frac{1}{2}$, then using the Cauchy-Schwarz inequality in the integral form, namely

$$|\int d\lambda f(\lambda)g(\lambda)|^2 \leqslant \int d\lambda |f(\lambda)|^2 \int d\lambda |g(\lambda)|^2 \tag{3}$$

and setting $f(\lambda) = \sqrt{p(\lambda)}$ and $g(\lambda) = \sqrt{p(\lambda)}\,\Sigma_m$, we obtain

$$\mathcal{E}_m \equiv |\langle \Sigma_m \rangle|^2 \leqslant \int d\lambda\, p(\lambda) \prod_{k=1}^{m} |\sigma_{s_k}^{(k)}(\lambda)|^2 = 2^{-m}. \tag{4}$$

Since this bound holds for all correlators as in Eq. (2), it is a many-body Bell inequality, and it will be our main tool for studying Bell correlations in complex systems throughout this work. For the original derivation of this inequality and its extended discussion, see [27, 29, 41]. The inequality from Eq. (4) is a member of a broader family of $N$-body Bell inequalities for qubits [42].

#### 2.1.2 Permutationally invariant case

Next, we assume that the $m$ parties are a subset of a larger set of $N$, $m \leqslant N$. Our aim is to derive a Bell inequality that would be invariant upon the permutation of any pair of parties. To this end, consider a correlator of $m$ outcomes

$$\langle \Sigma_{\vec{k}} \rangle = \left\langle \sigma_{s_{k_1}}^{(k_1)} \ldots \sigma_{s_{k_m}}^{(k_m)} \right\rangle. \tag{5}$$

Note that rather than labelling $\Sigma$ with a single number $m$, as in Eq. (1), we now use $\vec{k} = (k_1, \ldots, k_m)$, which denotes one particular choice of $m$ parties out of $N$. As previously, every party $k_i$ has a freedom of choice of the sign $s_{k_i}$. Next, we introduce a permutationally invariant correlator

$$\left\langle \tilde{\Sigma}_m \right\rangle = \left\langle \hat{\mathcal{S}}_{\vec{k}}^{(N)} \Sigma_{\vec{k}} \right\rangle = \left\langle \hat{\mathcal{S}}_{\vec{k}}^{(N)} \sigma_{s_{k_1}}^{(k_1)} \ldots \sigma_{s_{k_m}}^{(k_m)} \right\rangle. \tag{6}$$

The symmetrization operator $\hat{\mathcal{S}}_{\vec{k}}^{(N)}$ sums over all possible $\binom{N}{m}$ choices of $m$ parties out of $N$. Note that the symmetrization allows for all products $\sigma_{s_{k_1}}^{(k_1)} \ldots \sigma_{s_{k_m}}^{(k_m)}$, including those that are unordered. Hence, for every fixed $\vec{k}$ there are $m!$ equivalent correlators and the total number of permutations is $\|\hat{\mathcal{S}}_{\vec{k}}^{(N)}\| = \binom{N}{m} m! = \frac{N!}{(N-m)!}$. The Bell inequality now reads

$$\tilde{\mathcal{E}}_m = |\left\langle \tilde{\Sigma}_m \right\rangle|^2 = |\left\langle \hat{\mathcal{S}}_{\vec{k}}^{(N)} \Sigma_{\vec{k}} \right\rangle|^2 \leqslant \|\hat{\mathcal{S}}_{\vec{k}}^{(N)}\|^2 2^{-m} = \left( \frac{N!}{(N-m)!} \right)^2 2^{-m}. \tag{7}$$

### 2.1.3 More degrees of freedom

We now generalize the above findings to situations, where parties, besides yielding the binary outcomes, can be labeled with another degree(s) of freedom. A good illustration of such setup would be a collection of photons emitted in the spontaneous parametric down conversion (SPDC) process, where apart from the two orthogonal polarizations (binary results), emitted particles can occupy distinct momentum states. Another example is a collection of atomic pairs scattered / emitted from an $F = 1$, $m_F = 0$ Bose-Einstein condensate into states with opposite momenta in a spin-changing collision ($m_F = \pm 1$) [39].

Our aim is to adapt the correlator from Eq. (5) to account for these additional degrees of freedom. To this end we first consider the simplest possible case, namely parties marked with one label which is either $A$ or $B$—denoting two separate regions of space, distinct momenta, etc. We will address the question of extending these arguments to more complex configurations later on. The goal is to detect the Bell correlations in the system, but we focus on its particular form — such that extends over the regions. Thus we propose the correlator between $m$ parties in $A$ and $k$ in $B$ in the following form

$$\tilde{\mathcal{E}}_{m,k} = |\left\langle \tilde{\Sigma}_m^{(A)} \tilde{\Sigma}_k^{(B)} \right\rangle|^2, \tag{8}$$

where $\tilde{\Sigma}_{m/m}^{(A/B)}$ is defined in Eq. (6). According to Eq. (7), the Bell inequality now reads

$$\tilde{\mathcal{E}}_{m,k} = |\left\langle \tilde{\Sigma}_m^{(A)} \tilde{\Sigma}_k^{(B)} \right\rangle|^2 = |\left\langle \hat{\mathcal{S}}_{\vec{k}}^{(N_A)} \Sigma_{\vec{k}}^{(A)} \hat{\mathcal{S}}_{\vec{k}'}^{(N_B)} \Sigma_{\vec{k}'}^{(B)} \right\rangle|^2 \leqslant \|\hat{\mathcal{S}}_{\vec{k}}^{(N_A)}\|^2 \|\hat{\mathcal{S}}_{\vec{k}}^{(N_A)}\|^2 2^{-(m+k)}$$

$$= \left( \frac{N_A!}{(N_A - m)!} \frac{N_B!}{(N_B - m)!} \right)^2 2^{-(m+k)}, \tag{9}$$

where $N_A$ and $N_B$ is the number of particles per region.

Finally, we note that the approach outlined in the above paragraphs can be extended to more complicated situations where more regions / degrees of freedom are in play. To detect the Bell correlations between $\mu$ regions $A_1 \ldots A_\mu$ one could consider a correlator

$$\tilde{\mathcal{E}}_{m_1, \ldots m_\mu} = |\left\langle \tilde{\Sigma}_{m_1}^{(A_1)} \ldots \tilde{\Sigma}_{m_\mu}^{(A_\mu)} \right\rangle|^2, \tag{10}$$

and then proceed analogically to Eq. (9).

## 2.2 Testing with quantum systems

We now discuss in detail how these local realistic bounds can be tested in complex many-body systems of qubits.

### 2.2.1 Quantum: single degree of freedom

The inequality (4) is well-suited to test Bell correlations in many-body systems of individually addressable qubits, such as spin chains or optical lattices with one two-level atom per site. With $\hat{\sigma}_{s_k}^{(k)}$ being the Pauli rising ($s_k = +1$) / lowering ($s_k = -1$) operator for the $k$-th qubit, the correlator is

$$\mathcal{E}_m^{(q)} = |\langle \hat{\sigma}_{s_1}^{(1)} \dots \hat{\sigma}_{s_m}^{(m)} \rangle|^2, \tag{11}$$

where the symbol $\langle \cdot \rangle$ denotes the average calculated over a quantum state and the upper index $(q)$ indicates a quantum-mechanical correlator. Among states which violate the bound from Eq. (4), the GHZ state

$$|\psi\rangle = \frac{1}{\sqrt{2}} \left( |\uparrow\rangle^{\otimes N} + |\downarrow\rangle^{\otimes N} \right) \tag{12}$$

stands out, giving a maximal value of the correlator $\mathcal{E}_m^{(q)}$ for $m = N$ and either $s_k = 1$ or $s_k = -1$ for all $k$ simultaneously, for instance

$$\mathcal{E}_N^{(q)} = |\langle \hat{\sigma}_+^{(1)} \dots \hat{\sigma}_+^{(N)} \rangle|^2 = \frac{1}{4}. \tag{13}$$

On the other hand, the inequality (7) is tailored for bosonic systems of qubits, where particles cannot be addressed individually and occupy a single common mode (i.e., there are no other degrees of freedom apart from $|\uparrow\rangle$ and $|\downarrow\rangle$ for each particle), given that all $s_{k_i}$ are equal. The dimensionality of the Hilbert space reduces from $2^N$ to $N + 1$ and all the states can be represented in a form

$$\hat{\varrho} = \sum_{n,m=0}^{N} \varrho_{nm} |n, N-n\rangle\langle m, N-m|, \tag{14}$$

where $|n, N-n\rangle$ is a (symmetrized) state of $n$ qubits in $|\uparrow\rangle$ and $N - n$ in $|\downarrow\rangle$ while $\varrho_{nm}$ is the corresponding matrix element. Bosonic operators $\hat{a}^\dagger$ and $\hat{b}^\dagger$ create a single excitation in either $|\uparrow\rangle$ or $|\downarrow\rangle$ state and the symmetrized equivalent of the single-qubit rising / lowering operator $\hat{\sigma}_\pm^{(k)}$ is

$$\hat{J}_+ = \hat{a}^\dagger \hat{b} = \sum_{k=1}^{N} \hat{\sigma}_+^{(k)}, \tag{15a}$$

$$\hat{J}_- = \hat{a}\hat{b}^\dagger = \sum_{k=1}^{N} \hat{\sigma}_-^{(k)}. \tag{15b}$$

Since only the collective operations are now allowed, we must replace the product of $m$ operators addressing single qubits as in Eq. (11) with

$$\hat{\sigma}_{s_1}^{(1)} \dots \hat{\sigma}_{s_m}^{(m)} \longrightarrow \hat{J}_+^m \tag{16}$$

(or $\hat{J}_-^m$). Note that

$$\hat{J}_+^m = \left( \sum_{k=1}^{N} \hat{\sigma}_+^{(k)} \right)^m = \hat{\mathcal{S}}_{\vec{k}}^{(N)} \hat{\sigma}_+^{(k_1)} \dots \hat{\sigma}_+^{(k_m)}, \tag{17}$$

which is exactly a quantum equivalent of the averaged product in Eq. (6) with all $s_{k_i} = +1$. Hence the Bell inequality from Eq. (7) can be tested with quantum systems of bosonic qubits by seeking for violations of the following inequality

$$\tilde{\mathcal{E}}_m^{(q)} = |\langle \hat{J}_+^m \rangle|^2 \leqslant \left( \frac{N!}{(N-m)!} \right)^2 2^{-m}. \tag{18}$$

### 2.2.2 Quantum: more degrees of freedom

Analogically, the cross-region Bell correlations can be tested with the quantum correlator as below

$$\tilde{\mathcal{E}}_{m,k}^{(q)} = |\langle \hat{J}_+^{(A)m} \hat{J}_+^{(B)k} \rangle|^2, \tag{19}$$

where $\hat{J}_+^{(A/B)}$ are local operators for $A/B$. Using the Bell inequality from Eq. (8), a quantum system reveals Bell correlations if it violates

$$\tilde{\mathcal{E}}_{m,k}^{(q)} \leqslant \left( \frac{N_A!}{(N_A-m)!} \right)^2 \left( \frac{N_B!}{(N_B-k)!} \right)^2 2^{-(m+k)}. \tag{20}$$

However, does the breaking of the bound (20) imply the $A/B$ nonlocality? To answer this question, consider a following state

$$|\psi\rangle = \left[ \frac{|\uparrow\rangle^{\otimes N} + |\downarrow\rangle^{\otimes N}}{\sqrt{2}} \right]_A \otimes \left[ \frac{|\uparrow\rangle + |\downarrow\rangle}{\sqrt{2}} \right]_B \tag{21}$$

of $N$ qubits in $A$ forming the maximally entangled GHZ state and a single particle in $B$, uncorrelated with the rest, which is in a coherent superposition of $|\uparrow\rangle$ and $|\downarrow\rangle$. For this state, we obtain

$$\tilde{\mathcal{E}}_{N,1}^{(q)} = |\langle \hat{J}_+^{(A)N} \hat{J}_+^{(B)} \rangle|^2 = (N!)^2 2^{-4}. \tag{22}$$

This should be compared with the local realistic bound from Eq. (20) for $m = N$ and $k = 1$, which is $|\langle \hat{J}_+^{(A)N} \hat{J}_+^{(B)} \rangle|^2 \leqslant (N!)^2 2^{-(N+1)}$ and is violated by the right-hand side of Eq. (22) for $N > 3$. This violation is a result of the Bell correlations between the qubits located at $A$, rather than the $A/B$ nonlocality [43–45].

Thus one must use Eq. (20) with some precaution in order to detect the genuine interregion nonlocality: if the single-region correlators are Bell-bounded, i.e.,

$$\tilde{\mathcal{E}}_m^{(q)} = |\langle \hat{J}_+^{(A)m} \rangle|^2 \leqslant \left( \frac{N_A!}{(N_A-m)!} \right)^2 2^{-m}, \tag{23a}$$

$$\tilde{\mathcal{E}}_k^{(q)} = |\langle \hat{J}_+^{(B)k} \rangle|^2 \leqslant \left( \frac{N_B!}{(N_B-k)!} \right)^2 2^{-k}, \tag{23b}$$

and simultaneously the inequality from Eq. (20) is violated, then the non-locality stems from the cross-region correlations.

### 2.2.3 Quantum: non-fixed number of qubits

We now show how the correlator from Eq. (19) can detect the $A/B$ nonlocality in a system where the number of qubits is not fixed. To illustrate this point, we consider a process of

scattering of pairs of qubits into $A$ and $B$, with the perfect correlation in the $\uparrow / \downarrow$ degree of freedom, namely

$$|\psi_{AB}\rangle = \sum_{n,m=0}^{\infty} C_{n,m}|n:\uparrow\rangle_A|n:\downarrow\rangle_B|m:\downarrow\rangle_A|m:\uparrow\rangle_B \equiv \sum_{n,m=0}^{\infty} C_{n,m}|n,m\rangle_A|m,n\rangle_B, \qquad (24)$$

where the ket $|n,m\rangle_A$ denotes $n$ bosonic qubits in the $|\uparrow\rangle$ state and $m$ in $|\downarrow\rangle$, located in $A$ (and analogically for $B$). The coefficients $C_{n,m}$ are determined by the details of the scattering process.

To account for the qubit-number fluctuations, we split the sum over $n$ and $m$ in Eq. (24) in such a way so that first we introduce $N$, the number of particles in $A$ or $B$, then we sum over $n$, with $m = N - n$, and finally over all possible $N$'s, which gives

$$|\psi_{AB}\rangle = \sum_{N=0}^{\infty} \sum_{n=0}^{N} C_{n,N-n}|n,N-n\rangle_A|N-n,n\rangle_B \qquad (25)$$

$$= \sum_{N=0}^{\infty} \sqrt{p_N} \sum_{n=0}^{N} \mathcal{A}_n^N|n,N-n\rangle_A|N-n,n\rangle_B,$$

where we used the probability for having $N$ particles either in $A$ or $B$,

$$p_N = \sum_{n=0}^{N} |C_{n,N-n}|^2 \quad \text{and} \quad \mathcal{A}_n^N = \frac{C_{n,N-n}}{\sqrt{p_N}}. \qquad (26)$$

Since operators $\hat{J}_+^{(A/B)}$ do not locally modify the number of particles, superpositions between different $N$'s are not revealed by the correlator from Eq. (19). This means that in this context the state from Eq. (25) can be replaced by an incoherent mixture

$$\hat{\varrho}_{AB} = \sum_{N=0}^{\infty} p_N \hat{\varrho}_{AB}^{(N)}, \qquad (27)$$

where the fixed-$N$ density matrix is

$$\hat{\varrho}_{AB}^{(N)} = \sum_{n,m=0}^{N} \mathcal{A}_n^N \mathcal{A}_m^{N*}|n,N-n\rangle\langle m,N-m|_A \otimes |N-n,n\rangle\langle N-m,m|_B. \qquad (28)$$

The fact that the pure state from Eq. (25) is, from the point of view of the Bell test, equivalent with the mixture (27), is a manifestation of the superselection rules, which render coherences between states with different number of particles as nonphysical [46–50].

We now proceed to establish an upper bound for the correlator $\tilde{\mathcal{E}}_{m,k}^{(q)}$ in absence of non-local Bell correlations. Using Eqs (19), (27) and the Cauchy-Schwarz inequality, which in the discrete case takes the form of

$$|\sum_N a_N b_N|^2 \leqslant \sum_N |a_N|^2 \sum_N |b_N|^2 \qquad (29)$$

and setting $a_N = \sqrt{p_N}$ and $b_N = \sqrt{p_N} \text{Tr}\left[\hat{\varrho}_{AB}^{(N)}\hat{J}_+^{(A)m}\hat{J}_+^{(B)k}\right]$, we obtain

$$\tilde{\mathcal{E}}_{m,k}^{(q)} = |\sum_{N=0}^{\infty} p_N \text{Tr}\left[\hat{\varrho}_{AB}^{(N)}\hat{J}_+^{(A)m}\hat{J}_+^{(B)k}\right]|^2 \leqslant \sum_{N=0}^{\infty} p_N |\text{Tr}\left[\hat{\varrho}_{AB}^{(N)}\hat{J}_+^{(A)m}\hat{J}_+^{(B)k}\right]|^2$$

$$\equiv \sum_{N=0}^{\infty} p_N \tilde{\mathcal{E}}_{m,k}^{(q),N}. \qquad (30)$$

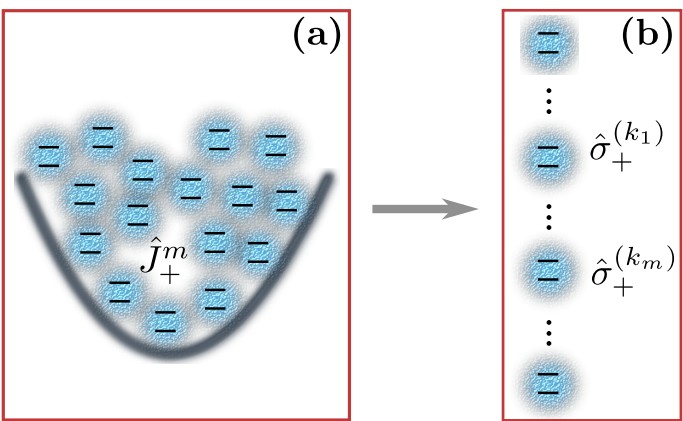

Figure 1: The test of Bell correlations in a bosonic systems. When **(a):** the qubits cannot be addressed individually, the Bell correlations are detected by means of a collective operator $\hat{J}_+$, which is replaced by local operators $\hat{\sigma}_+^{(k_i)}$ when **(b):** qubits are spatially separated.

This way, we upper-bounded $\tilde{\mathcal{E}}_{m,k}^{(q)}$, calculated with a full state (27) with a probabilistic combination of correlators $\tilde{\mathcal{E}}_{m,k}^{(q),N}$, where each such $\tilde{\mathcal{E}}_{m,k}^{(q),N}$ is calculated in a fixed-$N$ subspace. If in all fixed-$N$ sectors the correlators lie within the bound from Eq. (20), then the Bell nonlocality for the full state (27) is tested with

$$\tilde{\mathcal{E}}_{m,k}^{(q)} \leqslant 2^{-(m+k)} f_{mk}, \tag{31}$$

where

$$f_{mk} = \sum_{N=\max(m,k)}^{\infty} p_N \left( \frac{N!}{(N-m)!} \frac{N!}{(N-k)!} \right)^2. \tag{32}$$

If the inequality (31) is violated, then the system, as a whole, is non-local. Note however, that this criterion does not tell in which fixed-$N$ sectors the nonlocality resides. To answer this question, one would need to analyze $\tilde{\mathcal{E}}_{m,k}^{(q),N}$'s separately. We will further elaborate on this point in Section 3.2.

Though the argument presented in this Section was not entirely universal — we considered a particular non-fixed-$N$ state, see Eq. (24) — it is now clear how to proceed in general. One should consider the fixed-$N$ sectors of states in $A$ and $B$, for each determine the Bell bound and finally incoherently mix various contributions.

### 2.2.4 Additional remarks

A loophole-free Bell test requires the particles to be causally separated, so that the light cones associated with each of them do not intersect during the phase of local operations and measurements. However, for a collection of bosonic qubits occupying a single mode, the only available operations are the collective ones [see Fig. 1(a)], such as represented by the rising operator (15). This means that while the violation of Eq. (18) implies the presence of Bell correlations, to close the causality loophole these particles would need to be separated and addressed individually, as shown in Fig. 1(b). A similar approach, using collective spin operators to detect the Bell correlations in a bosonic system, has been experimentally tested in Ref. [51] (see also [52] for the theoretical background). While in [51] the two-body Bell correlations were witnessed in a spin-squeezed sample, the current work extends the analysis to the many-body correlations.

Note also that in a many-body system, like the Bose-Einstein condensate, it would be very hard to spatially split atoms apart without disturbing their quantum degrees of freedom. Nevertheless, the existing techniques, like the light-sheet method [53] or the detection of single metastable $^4$He atoms with micro-channel plates [39, 54] open the way for future complete Bell tests in systems of many-body bosonic qubits.

# 3 Applications

We now illustrate how the methods introduced in Section 2 apply to some prominent physical cases.

## 3.1 BEC in a double-well potential

First, we take a BEC in a double-well potential: an important example of a system of $N$ interacting ultra-cold bosonic qubits, governed by the two-mode Bose-Hubbard Hamiltonian,

$$\hat{H} = -\hat{J}_x + \frac{U}{N}\hat{J}_z^2. \tag{33}$$

Here, the collective operators $\hat{J}_x$ and $\hat{J}_z$ are two members of a triad

$$\hat{J}_x = \frac{1}{2}\sum_{i=1}^{n}\hat{\sigma}_x^{(i)} = \frac{1}{2}(\hat{a}^\dagger\hat{b} + \hat{a}\hat{b}^\dagger), \tag{34a}$$

$$\hat{J}_y = \frac{1}{2}\sum_{i=1}^{n}\hat{\sigma}_y^{(i)} = \frac{1}{2i}(\hat{a}^\dagger\hat{b} - \hat{a}\hat{b}^\dagger), \tag{34b}$$

$$\hat{J}_z = \frac{1}{2}\sum_{i=1}^{n}\hat{\sigma}_z^{(i)} = \frac{1}{2}(\hat{a}^\dagger\hat{a} - \hat{b}^\dagger\hat{b}) \tag{34c}$$

of the angular momentum operators satisfying the commutation relation $[\hat{J}_n, \hat{J}_m] = i\epsilon_{nmk}\hat{J}_k$, where $\epsilon_{nmk}$ is a matrix element of the Levi-Civita tensor (for brevity, we used the Einstein summation convention here). The dimensionless parameter $U$ in Eq. (33) determines the strengths of the two-body interactions in units of the Josephson energy [25].

We focus on the negative-$U$ case (attractive interactions) and find the ground state of the Hamiltonian (33) with $N = 100$ qubits. This is calculated by first fixing a value of $U$ and then by performing the exact diagonalization of the above Hamiltonian decomposed in the basis of the eigen-states of $\hat{J}_z$, see Eq. (14). First, we scrutinize the vicinity of $U = -1$, since at this point (in the limit of $N \to \infty$) the system undergoes a quantum phase transition [31, 55–59]. The significance of this point is underlined by the observation that upon passing $U = -1$, the correlators $\tilde{\mathcal{E}}_m^{(q)}$ [here shown for $m = 70, 80, 90$ and $100$ and normalized to the right-hand-side of Eq. (18)], cross the limit and start to witness the Bell correlations, see Fig. 2. This is in line with prior works [55, 60].

When $U$ drops below this critical point, the system enters a deeply quantum regime [60, 61], where the ground state of the Hamiltonian (33) is a macroscopic superposition of two largely occupied states and tends to the $N00N$ state, i.e.,

$$|\psi\rangle \xrightarrow{U \to -\infty} \frac{1}{\sqrt{2}}\Big(|N, 0\rangle + |0, N\rangle\Big). \tag{35}$$

For this maximally entangled state, the quantum features are present only in the highest-order correlation function, thus only the full $N$-body correlator is non-vanishing and reaches

a maximal value

$$\tilde{\mathcal{E}}_N^{(q)} = \frac{1}{4}(N!)^2 \, . \tag{36}$$

Figure 3 shows the same correlators ($m = 70, 80, 90, 100$) but in a wider range of interaction strengths, $U \in [-50, 0]$. Clearly, at highly negative $U$, only the maximal correlator remains above the local realistic bound.

## 3.2 Four-mode SPDC

Next, our focus is on the $A/B$ configuration, as discussed in Section 2.2.2. We consider a broad family of scattering processes (SPDC, emissions of atomic pairs from a BEC, etc.), governed by a Hamiltonian (in units of coupling energy $g$)

$$\hat{H} = \hat{a}_\uparrow^\dagger \hat{b}_\downarrow^\dagger + \hat{a}_\downarrow^\dagger \hat{b}_\uparrow + \text{h.c.} \, . \tag{37}$$

The mode-functions associated with operators $\hat{a}_{\uparrow/\downarrow}$ are localized in region $A$, and analogically for $\hat{b}_{\uparrow/\downarrow}$ and $B$. Since the two parts of the Hamiltonian from Eq. (37), $\hat{a}_\uparrow^\dagger \hat{b}_\downarrow^\dagger$ and $\hat{a}_\downarrow^\dagger \hat{b}_\uparrow$, commute, the coefficient from Eq. (24) factorizes, $C_{nm} = C_n C_m$ , with

$$C_n = \frac{1}{\cosh(t)} \left(-i \tanh(t)\right)^n \, . \tag{38}$$

The resulting state is

$$|\psi\rangle = \sum_{n,m} \frac{(-i \tanh(t))^{n+m}}{\cosh^2(t)} |n, m\rangle_A \otimes |m, n\rangle_B \, . \tag{39}$$

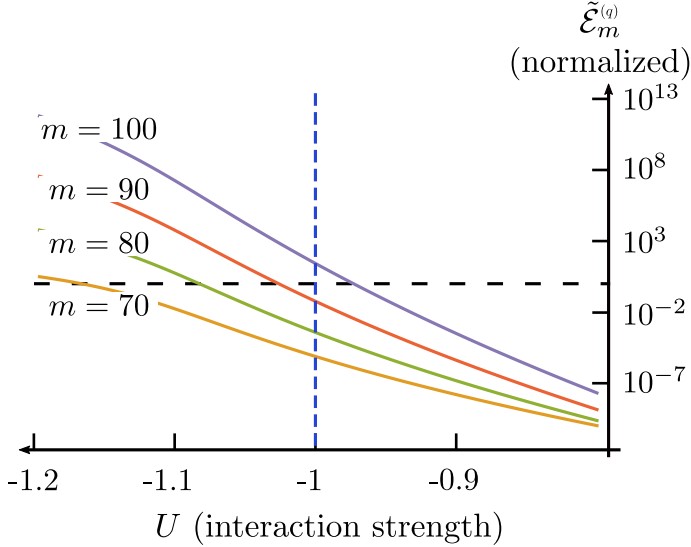

Figure 2: The Bell correlators $\tilde{\mathcal{E}}_m^{(q)}$ calculated with the ground state of the Hamiltonian from Eq. (33) with $N = 100$ qubits. The correlators are drawn as a function of the interaction strength $U$ in the vicinity of the critical point $U = -1$ (marked with a vertical dashed blue line), for $m = 70, 80, 90$ and $100$. The correlators are normalized to the right-hand side of inequality (18), so that the dashed horizontal line at a constant value $\tilde{\mathcal{E}}_m^{(q)} = 1$ marks the local realistic bound for all $m$.

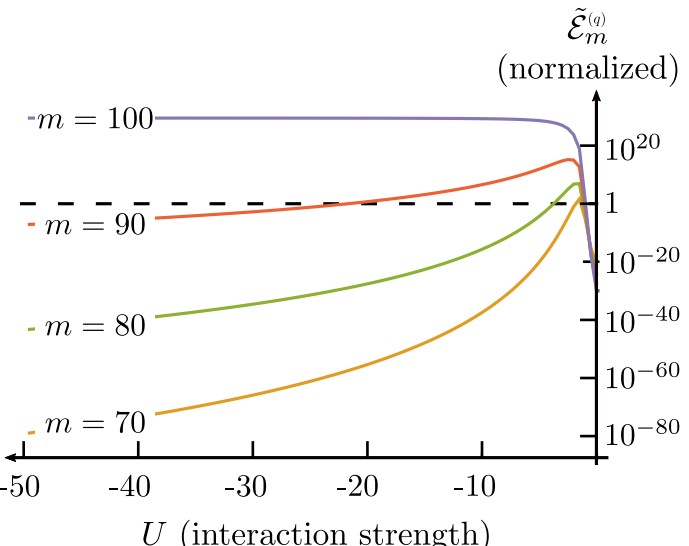

Figure 3: The Bell correlators $\tilde{\mathcal{E}}_m^{(q)}$ calculated with the ground state of the Hamiltonian (33) with $N = 100$ qubits, as a function of the interaction strength $U \in [-50, 0]$, and for $m = 70, 80, 90$ and $100$. The correlators are normalized as in Fig. 2.

In the Heisenberg picture, we have

$$\hat{a}_\uparrow(t) = \cosh(t)\hat{a}_\uparrow(0) - i \sinh(t)\hat{b}_\downarrow^\dagger(0) \tag{40}$$

and analogically for the remaining three operators. Due to the $A/B$ symmetry of the Hamiltonian (37), we consider the correlator $\tilde{\mathcal{E}}_{m,k}^{(q)}$ with $k = m$, and using Eq. (40) we obtain

$$\tilde{\mathcal{E}}_{m,m}^{(q)} = |\langle \hat{J}_+^{(A)m} \hat{J}_-^{(B)m} \rangle|^2 = (\sinh(t)\cosh(t))^{4m}(m!)^4. \tag{41}$$

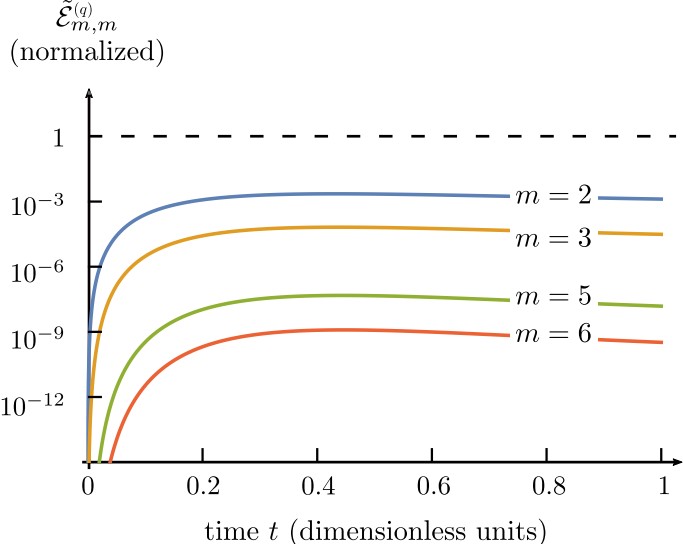

Figure 4: The correlator $\tilde{\mathcal{E}}_{m,m}^{(q)}$ from Eq. (41) for $m = 2, 3, 5, 6$ (top to bottom curve) as a function of time and normalized to the local realistic bound (31). The Figure shows that the values of all the correlators lie well below the Bell limit.

Next, we establish the local realistic bound, using the approach detailed in Section 2.2.2. To this end, we express the state (39) as follows

$$|\psi\rangle = \sum_{N=0}^{\infty} (-i)^N \sqrt{p_N} |\psi_N\rangle, \tag{42}$$

where the fixed-$N$ state is

$$|\psi_N\rangle = \sum_{n=0}^{N} \frac{|n, N-n\rangle_A \otimes |N-n, n\rangle_B}{\sqrt{N+1}}, \tag{43}$$

and $p_N = \frac{\tanh^{2N}(t)}{\cosh^4(t)}(N+1)$ is the probability for having $N$ particles in $A/B$, while $N = n + m$. Since local operators $\hat{J}_+^{(A)}/\hat{J}_-^{(B)}$ conserve the number of particles, this state can be replaced by an incoherent mixture

$$\hat{\varrho} = \sum_{N=0}^{\infty} p_N |\psi_N\rangle\langle\psi_N|. \tag{44}$$

Once the $p_N$ is determined, the local realistic bound is given by Eq. (31) with $m = k$, i.e.,

$$
\begin{aligned}
f_{mm} &= (m!)^4 \sum_{N=m}^{\infty} p_N \binom{N}{m}^4 \tag{45} \\
&= \frac{(m!)^4}{\cosh^2(t)} \left( \binom{0}{m}^4 {}_5F_4\left[\vec{a}_1, \vec{m}_1, \tanh^2(t)\right] + \binom{1}{m}^4 {}_5F_4\left[\vec{a}_2, \vec{m}_2, \tanh^2(t)\right] \tanh^2(t) \right),
\end{aligned}
$$

where $_5F_4$ stands for a generalized hypergeometric function and

$$\vec{a}_i = (i, i, i, i, i)^{\mathrm{T}}, \tag{46a}$$

$$\vec{m}_i = (i-m, i-m, i-m, i-m)^{\mathrm{T}}. \tag{46b}$$

In Fig. 4 we plot the $\tilde{\mathcal{E}}_{m,m}^{(q)}$ normalized to the local realistic bound, i.e., divided by $f_{mm} 2^{-2m}$ for $m = 2, 3, 5$ and 6, as a function of time. Clearly, all these correlators lie well below the bound, thus neither of them detects any Bell correlations.

This however does not mean that the state from Eq. (39) contains no highly non-classical correlations, though we claim that they reside in the fixed-$N$ subspaces and the corresponding states $|\psi_N\rangle$ defined in Eq. (43). To show this, we calculate

$$\tilde{\mathcal{E}}_{m,m}^{(q),N} = |\langle\psi_N|\hat{J}_+^{(A)m}\hat{J}_-^{(B)m}|\psi_N\rangle|^2 \tag{47}$$

and plot this correlator normalized to the Bell bound, i.e., the right-hand side of the Bell inequality

$$\tilde{\mathcal{E}}_{m,m}^{(q),N} \leqslant \left(\frac{N!}{(N-m)!}\right)^4 2^{-2m}, \tag{48}$$

for $N = 2, 3, 6$ and 12 and for each $N$ changing $m$ from 1 to $N$, see Fig. 5. Clearly, in fixed-$N$ subspaces, the many-body Bell correlations is present in the state (43) and it is genuinely a result of the cross-region nonlocality. In a single region alone, say $A$, the qubits are uncorrelated because the reduced state is a full mixture

$$\hat{\varrho}_N^{(A)} = \mathrm{Tr}[|\psi_N\rangle\langle\psi_N|]_B = \sum_{n=0}^{N} \frac{|n, N-n\rangle\langle n, N-n|_A}{N+1} = \frac{1}{N+1}\hat{\mathbb{1}}_A, \tag{49}$$

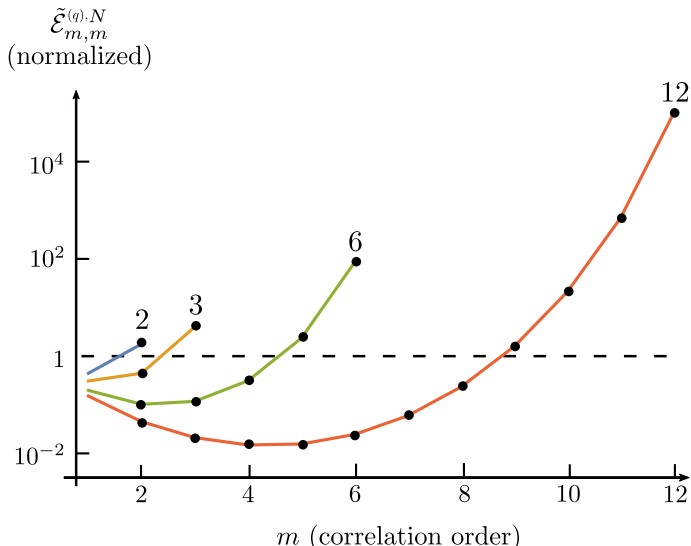

Figure 5: The correlator from Eq. (47) calculated with the fixed-$N$ part [see Eq. (43)] of the full state [Eq. (39)] for $N = 2, 3, 6$ and $12$, as a function of $m \in [1, \dots N]$ and normalized to the local realistic bound from Eq. (48). Contrary to the correlator $\tilde{\mathcal{E}}_{m,m}^{(q)}$ calculated with a full state, which does not break the Bell inequality with $m = 2, 3, 5$ and $6$, here the bound is surpassed (for instance $N = 6$ and $m = 5, 6$).

which gives $\mathrm{Tr}\left[ \hat{J}_{+}^{(A)m} \hat{\varrho}_{N}^{(A)} \right] = 0$ for all $m > 0$. Symmetrically, the same argument holds for the region $B$.

By inspecting the Fig. 5 we notice that already at the lowest $N = 2$ (i.e., two particles per $A/B$), the correlator $\tilde{\mathcal{E}}_{2,2}^{(q),2}$ breaks the local realistic bound. However, for larger $N$, the $\tilde{\mathcal{E}}_{2,2}^{(q),N}$ remains below the bound.

This explains why the Bell nonlocality is not detected in the full state from Eq. (39). For various $N$'s, the nonlocality is witnessed by the correlators of different orders $m$. Hence the mixing of contributions from all $N$'s but with fixed $m$ adds together a few contributions $\tilde{\mathcal{E}}_{m,m}^{(q),N}$ that detect the nonlocality with a majority that lie below the Bell bound. In consequence, the averaged correlator $\tilde{\mathcal{E}}_{m,m}^{(q)}$ does not break the nonlocality limit. The conclusion from this Section is that the nonlocality can be detected in such an $A/B$ configuration following either of two paths. First is the standard approach, where one restricts to a very low-gain regime, where the probability of having more than a single qubit per region is negligible. In such case, the CHSH inequality will detect the Bell correlations. The second is the many-body regime, and in this case one should apply the post-selection, restrict to the fixed-$N$ subspace and use the correlators $\tilde{\mathcal{E}}_{m,m}^{(q),N}$. Otherwise, the mixing of different $N$'s, as in Eq. (44), washes out any multiparticle non-local effects. Note that the influence of fluctuations of the number of particles on the strength of the Bell correlations has been widely discussed in [51].

# 4 Discussion

Before we summarize, let us take a closer look at the correlator $\tilde{\mathcal{E}}_{m}^{(q)}$ from Eq. (18) for $m = 3$. Using $\hat{J}_{+} = \hat{J}_{x} + i\hat{J}_{y}$ we obtain

$$\tilde{\mathcal{E}}_{3}^{(q)} = |\langle \hat{J}_{+}^{3} \rangle|^{2} = \left( \left\langle \hat{J}_{x}^{3} - \hat{J}_{y}\hat{J}_{x}\hat{J}_{y} - \hat{J}_{y}^{2}\hat{J}_{x} - \hat{J}_{x}\hat{J}_{y}^{2} \right\rangle \right)^{2} + \left( \left\langle \hat{J}_{y}^{3} - \hat{J}_{x}\hat{J}_{y}\hat{J}_{x} - \hat{J}_{x}^{2}\hat{J}_{y} - \hat{J}_{y}\hat{J}_{x}^{2} \right\rangle \right)^{2}. \quad (50)$$

In the experiment, to know $\tilde{\mathcal{E}}_3^{(q)}$ is to measure $2^3 = 8$ different 3-rd order correlation functions (like $\langle \hat{J}_x^2 \hat{J}_y \rangle$), which generalizes to $2^m$ $m$-th order functions for $\tilde{\mathcal{E}}_m^{(q)}$. This is very challenging already for $m = 3$ and the possibility to empirically evaluate the Bell correlator for high $m$'s is a rather distant perspective.

Nevertheless, the experimental progress in single-particle detection is enormous. For photons, such devices are of almost every-day use in laboratories. For atoms, various techniques have been developed, including the aforementioned light-sheet method [53] or the luminescence from atoms trapped in an optical lattice [62, 63]. Last but not least, spatial correlations of up to 6-th order between meta-stable $^4$He atoms have been measured with ultra-high accuracy [54].

This work is intended mostly as a theoretical study of Bell correlations in many-body bosonic configurations. Such inquiry brings forward the fundamental aspects of complex systems. Nevertheless, there is also an application-oriented flavor of this inquiry, because the many-body Bell correlations, just as the entanglement [64] and the EPR steering [65], is a resource for the quantum-enhanced metrology [31, 66].

Finally, we would like to bring up the relation between the $\tilde{\mathcal{E}}_m^{(q)}$ and those other (than the Bell-type) quantum correlations: the entanglement and the EPR steering. For spin-chains, when qubits can be addressed individually, it is justified to use the correlator $\mathcal{E}_m^{(q)}$ from Eq. (11). It detects entanglement when $\mathcal{E}_m^{(q)} > 4^{-m}$ and Bell correlations if $\mathcal{E}_m^{(q)} > 2^{-m}$ [29,30]. Between these two threshold, lies a border, above which $\mathcal{E}_m^{(q)}$ detects the EPR steering, and its position depends on the physical setup—namely what is the ratio between the active (steering) and the passive (being steered) qubits [27]. The limits attributed to $\mathcal{E}_m^{(q)}$ can be translated into limits for $\tilde{\mathcal{E}}_m^{(q)}$ like in Eq. (18), allowing for an even deeper and more systematic study of quantum correlations in many-body bosonic systems.

Note also that the GHZ (12), (21) and the N00N states (35) are considered in this work merely for illustration as the limiting cases to establish the bounds for the witnesses of Bell correlations in quantum systems. Naturally, these maximally-entangled states are mostly vulnerable to particle losses and for higher $N$ should not be regarded as realistic testing tools for the theory.

# 5 Conclusions

We studied the many-body Bell correlations in bosonic systems. To this end, we adopted a Bell correlator from the case when qubits can be addressed individually to the situation when only collective bosonic operations are allowed. Next, we showed how to construct a witness of Bell correlations for even more complex scenarios, when qubits, apart from their internal two-level structure, can occupy a number of external modes. Afterwards, we generalized these results to systems, where the number of particles is not fixed.

With all these tools at hand, we considered a number of examples, such as a Bose-Einstein condensate in a double-well potential with attractive interactions or a collection of photons obtained in a spontaneous parametric-down conversion process. While in the former case, the many-body Bell correlations naturally emerged in the ground state, for the latter a more detailed analysis was necessary. This was due to the fact that the full state contained a non-fixed number of photons. We showed that strong Bell correlations reside in fixed-$N$ sectors, rather then in the photonic state taken as a whole.

The presented method allows for a systematic analysis of all the members of the triad of quantum correlations: entanglement, EPR steering and Bell correlations. We hope that such study expands the knowledge on highly non-classical effects in many-body systems. This

has implications both for the basic research but also for the application-oriented quantum technologies.

## Acknowledgements

The support of the National Science Centre, Poland under the QuantERA programme, Project no. 2017/25/Z/ST2/03039 is acknowledged.

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
