# Peer review of "Many-body Bell inequalities for bosonic qubits"

_SciPost Physics Core, doi:SciPost Phys. Core 5, 025 (2022)_

## Round 1 · Referee Report · Anonymous (Referee 2) · 2022-1-4

Strengths

1- This is a study about Bell correlation witnesses in bosonic systems, showcasing interesting tools for the study of high-order correlator functions.
2- The paper tackles some hard questions such as the ones of Bell correlations across two space-like separated regions (double-well BEC), giving elegant analytical results.
3- The paper proposes interesting physically relevant situations in which bosonic systems could potentially reveal Bell correlations, and discusses the challenges towards their eventual detection in an experiment.

Weaknesses

1- The work's motivation is the study of Bell's nonlocality in bosonic qubits, highly emphasized through examples of Bose-Einstein condensates, where individual addressing is manifestly not possible. In my view, the manuscript glosses over the issue of space-like separation too much, so that it may be confusing to the reader.
2- Another item that is confusing is that the paper uses interchangeably the domain of quantum physics, with qubits and bosons and operators acting on Hilbert or Fock spaces with that of Bell inequalities, which talk about correlations and probability distributions. This is a key difference in talking about entanglement vs Bell nonlocality and it should be made clear in the paper.
3- The study is restricted to the so-called Bell correlation witnesses (strictly speaking, there is no single Bell inequality with a classical bound, inputs and outputs in a black box scenario, etc.). Although duly acknowledged, some of the examples are almost impossible to implement experimentally, due to the non-robustness of particle losses.
4- One of the highlights is that stronger Bell correlations reside in fixed N sectors, but that is not very surprising. For instance, in [40] it was shown how the operators in the Bell correlation witness commute with the particle number operator.

Report

My recommendation for the paper is a major revision. The idea and the motivation are good and the math seems technically correct. However, the weaknesses I highlighted should be either resolved or explicitly acknowledged in the paper.

Given the originality and the advancement of the current state of knowledge, I think this work should be eventually publishable in SciPost Physics Core, and does not transcend its criteria significantly enough to be more appropriate for a more selective journal.

Requested changes

1- Before Eq. 2, lambda is a "hidden variable", but Bell's theorem is about local hidden variables. The issue of locality is key here, especially when dealing with indistinguishable particles. How does Bell's theorem formulate in this case? What are the measurement choices in the Bell experiment, since there appears only to be the \sigma_plus observable for each particle. That connection should be made explicit, beyond what Fig. 1 hints at.
2- k \in [1,m] is perhaps better to say k \in \{1,\ldots,m\}
3- The Cauchy-Schwarz inequality in integral form is used repeatedly in the paper. To make it more self-contained I would suggest to state it in general form the first time it is used.
4- In the 3.1 example, it would be nice if more details about how the ground state for N=100 was found were indicated.
5- In Figure 1, what would be the physical mechanism by which one would space-like separate the particles in a BEC without changing their (relevant) quantum degrees of freedom?

References:
R1- In the introduction, two more loophole-free Bell tests were performed in 2015 almost simultaneously with the one cited (and another in 2017) by independent groups.
R2- In the discussion about Eq. 18 the problem about nonlocality depth is highlighted. There exist works that already address this problem in the context of device-independent witnesses of entanglement depth that should be referenced as well.
R3- The triplet of experiments on split BEC https://www.science.org/doi/abs/10.1126/science.aat4590 seems also a good motivation for this paper worthy of mentioning.
R4- The entanglement witness bound 4^{-m} in Section 4 needs proper referencing.

---

## Round 1 · Referee Report · Anonymous (Referee 3) · 2022-1-25

Report

The main aim of this work is to construct Bell or Bell-type inequalities for bosonic many-qubit states. First, the Author considers a simple Bell inequality, constructed for instance in Ref. [24] for a pair of binary observables per party, and derives a symmetric version of it in which all observers measure the same observables. Second, the Author considers a situation in which a cloud of particles is divided into two groups and, building on the above symmetric inequality, he derives another inequality, symmetrized within each group, whose aim is to detect nonlocal correlations between these groups. A highly interesting feature of these constructions is that the obtained inequalities can be rewritten in terms of expectation values of some collective quantities and their powers and thus in principle could be used to detect Bell correlations within current experiments; an example of such an experiment was recently performed in Ref. [40]. Finally, the obtained inequalities are tested on two particular physical systems, for instance the BEC condesate in a double-well potential.

While I understand the general aim of this work and find it certainly interesting, I have certain doubts about the approach used here to achieve this goal:

1. While the inequality (3) seems to be derived for arbitrary pairs of binary observables measured by the parties, which might also differ between different parties, in the ‘quantum version’ of it, stated in (4), the Author seems to assume that all observers measure the same particular observables which are the Pauli matrices sigma_x and sigma_y. This is not a serious problem because at the end of the day one always needs to choose a particular Bell operator to test whether a given state is non-local or not. Here, however, the passage from (3) to (4) is made without basically any explanation, and it would make the paper easier to follow if the Author elaborated on this step. It should be noticed here that (3) is a Bell inequality in which also in the quantum regime every party can in principle measure any pair of observables.

2. A slightly more serious problem is with the inequality (13). To derive it the Author starts from (10) which is an expectation value of a particular quantum operator being simply a symmetrized version of (4). This is certainly not a proper way to derive Bell inequalities. And, even if this procedure could lead to a valid Bell inequality because it relies on (3), in my opinion this part of the paper is not properly phrased. In particular, I would first introduce a symmetrized version of the Bell expression (3), that is, one which is invariant under permutation of any pair of parties; let me add here that the fact that a Bell expression is symmetric does not impose any constraints on the observables that the parties measure. Now, by fixing all the observables to be sigma_x and sigma_y, one would recover the expectation value in (10) or (13).

Another possibility is that the Author simply derives a permutationally-invariant version of the particular Bell operator given in (4) in order to rewrite it in terms of powers of collective observables. But then, I am not sure whether it’s entirely correct to name (13) Bell inequality. If it’s the case, I suggest to add a few more lines to elaborate on what exactly is being done there.

3. A similar problem is with (21) or (23). Here, the Author again considers an expectation value of a particular quantum operator being a tensor product of two sequences of the rising operators of different lengths. Then, in Eq. (21) he attempts to put an upper bound on it for LHV models by using Eq. (6), which, I believe, is the maximal expectation value of a tensor product of such a sequence of the rising operators. However, I don’t think that this procedure leads to a proper Bell inequality because to obtain a Bell inequality one needs to optimize over all possible LHV models. For instance, assuming that each party decides to measure two same observables sigma_x (which is a valid possibility in a Bell scenario), the expectation value of (6) will be larger than ¼. It seems that what the Author derives is rather an entanglement witness. I thus suggest to modify this part of the paper accordingly.

Concluding, in my opinion the paper presents a very interesting line of research that merits publication, however, prior to that I suggest to revise the manuscript significantly, in particular taking into account the above comments.

Other comments:

1. ‘a simplest case’ → ‘the simplest case’.
2. Eq. (25), ‘n.m=0’ → ‘n,m=0’.
3. The Author uses ‘nonlocality bound’ to name the maximal value of (3) or other expressions over the LHV models. Usually one uses the ‘LHV bound’ or ‘local realistic bound’ for that purpose.

---

## Round 2 · Referee Report · Anonymous (Referee 2) · 2022-3-7

Strengths

See previous report

Weaknesses

See previous report

Report

I have checked the changes in the revised version and I am satisfied in the way my previous concerns have been addressed. I think the paper has overall improved in quality and clarity. Therefore, I am happy to recommend the paper for publication.

Requested changes

Please fix typos: "permuationally", Eq. 9 should end with a comma, etc.

---

## Round 2 · Referee Report · Anonymous (Referee 1) · 2022-3-30

Report

Let me first apologize for keeping the paper for so long.

I am very happy to see that the Author took into account all my comments when preparing the new version of the manuscript. I think that now the paper can be recommended for publication. I would nevertheless have one comment concerning the terminology. Namely, I am not sure whether it’s entirely correct to use Bell nonlocality to term the correlations observed in a bosonic system in which individual particles cannot be addressed. It’s true that they violate a Bell inequality, however, the experiment leading to those correlations is not a proper Bell experiment. For this reason Ref. [53] proposes to call correlations of this type Bell correlations, and I suggest to also use this terminology in the present work.

---

## Round 2 · Author Response

\documentclass[aps,pra,onecolumn,notitlepage]{revtex4-1} \usepackage{amsmath,amssymb,graphicx} \usepackage[T1]{fontenc}

\begin{document}

\newcommand{\sep}{\bigskip \begin{center} ---------$\circ$--------- \end{center} \bigskip }

\title{Response to the Referees} \maketitle

Dear Prof. T\'oth,

\bigskip I would like to thank you for the handling of this manuscript and the Referees for their insightful reports. I hope that after all the modifications following the Referees suggestions and remarks, and in line with their positive comments, the manuscript is now suitable for publication in the SciPost Physics Core.

\bigskip

With kind regards,

Jan Chwede\'nczuk \bigskip \bigskip \bigskip

\noindent\makebox[\linewidth]{\rule{18cm}{0.4pt}} \noindent

\noindent\makebox[\linewidth]{\rule{18cm}{0.4pt}}

\bigskip \bigskip

\noindent {\bf Comment I.1}: {\it While the inequality (3) seems to be derived for arbitrary pairs of binary observables measured by the parties, which might also differ between different parties, in the ‘quantum version’ of it, stated in (4), the Author seems to assume that all observers measure the same particular observables which are the Pauli matrices $\sigma_x$ and $\sigma_y$. This is not a serious problem because at the end of the day one always needs to choose a particular Bell operator to test whether a given state is non-local or not. Here, however, the passage from (3) to (4) is made without basically any explanation, and it would make the paper easier to follow if the Author elaborated on this step. It should be noticed here that (3) is a Bell inequality in which also in the quantum regime every party can in principle measure any pair of observables. }

\bigskip

\noindent {\bf Response}: I admit that this passage was not sufficiently clear and the sudden loss of generality was not properly discussed. It this new version, I allowed for the settings to be individually chosen by each party up until discussing the bosonic part where the nonlocality is tested with quantum systems of bosonic qubits.

\bigskip \sep \bigskip

\noindent {\bf Comment I.2}: {\it A slightly more serious problem is with the inequality (13). To derive it the Author starts from (10) which is an expectation value of a particular quantum operator being simply a symmetrized version of (4). This is certainly not a proper way to derive Bell inequalities. And, even if this procedure could lead to a valid Bell inequality because it relies on (3), in my opinion this part of the paper is not properly phrased. In particular, I would first introduce a symmetrized version of the Bell expression (3), that is, one which is invariant under permutation of any pair of parties; let me add here that the fact that a Bell expression is symmetric does not impose any constraints on the observables that the parties measure. Now, by fixing all the observables to be $\sigma_x$ and $\sigma_y$, one would recover the expectation value in (10) or (13).

Another possibility is that the Author simply derives a permutationally-invariant version of the particular Bell operator given in (4) in order to rewrite it in terms of powers of collective observables. But then, I am not sure whether it’s entirely correct to name (13) Bell inequality. If it’s the case, I suggest to add a few more lines to elaborate on what exactly is being done there. }

\bigskip

\noindent {\bf Response}: I would like to particularly thank for this observation. This was indeed an important flaw of the manuscript in the version reviewed by the Referees. Following the Referee's suggestion I have significantly revised the manuscript. In particular, the Section 2 now consists of two parts. The first one (2.1) deals with the problem of establishing the necessary set of Bell inequalities while the second (2.2) is strictly devoted to testing these inequalities with quantum systems. This way, I hope I managed to clearly separate the most general discussion at the level of probabilities from the realm of quantum mechanics.

\bigskip \sep \bigskip

\noindent {\bf Comment I.3}: {\it A similar problem is with (21) or (23). Here, the Author again considers an expectation value of a particular quantum operator being a tensor product of two sequences of the rising operators of different lengths. Then, in Eq. (21) he attempts to put an upper bound on it for LHV models by using Eq. (6), which, I believe, is the maximal expectation value of a tensor product of such a sequence of the rising operators. However, I don’t think that this procedure leads to a proper Bell inequality because to obtain a Bell inequality one needs to optimize over all possible LHV models. For instance, assuming that each party decides to measure two same observables $\sigma_x$ (which is a valid possibility in a Bell scenario), the expectation value of (6) will be larger than $1/4$. It seems that what the Author derives is rather an entanglement witness. I thus suggest to modify this part of the paper accordingly.}

\bigskip

\noindent {\bf Response}: I would like to thank again for the careful reading of the manuscript and for spotting this inconsequence. Indeed, the assumption of $1/4$ for each sub-region was taken from quantum-mechanical considerations for the GHZ state. As this paragraph had no connection with the remaining part of the manuscript, I decided to remove it from its current version in order to avoid any confusion by suddenly switching to the discussion of the cross-region entanglement rather than the nonlocality itself.

\bigskip \sep \bigskip

\noindent {\bf Comment I.4}: {\it Concluding, in my opinion the paper presents a very interesting line of research that merits publication, however, prior to that I suggest to revise the manuscript significantly, in particular taking into account the above comments. }

\bigskip

\noindent {\bf Response}: I would like to thank for this positive assessment of my work. I hope that after these substantial revisions the manuscript deserves publication in the SciPost Physics Core.

\bigskip \sep \bigskip

\noindent {\bf Comment I.5}: {\it Other comments:

  1. ‘a simplest case’ $\rightarrow$ ‘the simplest case’.

  2. Eq. (25), ‘n.m=0’ $\rightarrow$ ‘n,m=0’.

  3. The Author uses ‘nonlocality bound’ to name the maximal value of (3) or other expressions over the LHV models. Usually one uses the ‘LHV bound’ or ‘local realistic bound’ for that purpose.}

\bigskip

\noindent {\bf Response}: All these points have been addressed in the resubmitted version of the manuscript.

\bigskip \bigskip \bigskip

\noindent\makebox[\linewidth]{\rule{18cm}{0.4pt}} \noindent

\noindent\makebox[\linewidth]{\rule{18cm}{0.4pt}}

\bigskip \bigskip

\noindent {\bf Comment II.1}: {\it

1- This is a study about Bell correlation witnesses in bosonic systems, showcasing interesting tools for the study of high-order correlator functions.

2- The paper tackles some hard questions such as the ones of Bell correlations across two space-like separated regions (double-well BEC), giving elegant analytical results.

3- The paper proposes interesting physically relevant situations in which bosonic systems could potentially reveal Bell correlations, and discusses the challenges towards their eventual detection in an experiment.}

\bigskip

\noindent {\bf Response}: I would like to thank the Referee for their approbative report and all the constructive remarks. \bigskip

\bigskip \sep \bigskip

\noindent {\bf Comment II.2}: {\it The work's motivation is the study of Bell's nonlocality in bosonic qubits, highly emphasized through examples of Bose-Einstein condensates, where individual addressing is manifestly not possible. In my view, the manuscript glosses over the issue of space-like separation too much, so that it may be confusing to the reader. }

\bigskip

\noindent {\bf Response}: I admit that there might have been too much emphasis put on the spatial separation problem. I have shortened the remark around Fig. 1 but added a paragraph discussing the difficulty in spatially separating and addressing individual atoms from a BEC without disturbing their internal states (see also the answer to Comment II.10).

\bigskip

\bigskip \sep \bigskip

\noindent {\bf Comment II.3}: {\it Another item that is confusing is that the paper uses interchangeably the domain of quantum physics, with qubits and bosons and operators acting on Hilbert or Fock spaces with that of Bell inequalities, which talk about correlations and probability distributions. This is a key difference in talking about entanglement vs Bell nonlocality and it should be made clear in the paper.}

\bigskip

\noindent {\bf Response}: This indeed has been a substantial flaw of the previous version of the manuscript. I have completely reworked the Section 2 of the manuscript and split it into the discussion of the general problem of Bell nonlocality and separately into the part discussing the testing of these inequalities with quantum systems (see also my answer to Comment I.2).

\bigskip

\bigskip \sep \bigskip

\noindent {\bf Comment II.4}: {\it The study is restricted to the so-called Bell correlation witnesses (strictly speaking, there is no single Bell inequality with a classical bound, inputs and outputs in a black box scenario, etc.). Although duly acknowledged, some of the examples are almost impossible to implement experimentally, due to the non-robustness of particle losses.}

\bigskip

\noindent {\bf Response}: I agree that some of the examples, like the GHZ or the NOON state are rather text-book examples of most nonclassical states, since a loss of a single particle reduces these states into fully classical mixtures. I have addressed this problem in this reworked manuscript, see Section 4 (Discussion).

\bigskip

\bigskip \sep \bigskip

\noindent {\bf Comment II.5}: {\it One of the highlights is that stronger Bell correlations reside in fixed N sectors, but that is not very surprising. For instance, in [40] it was shown how the operators in the Bell correlation witness commute with the particle number operator.}

\bigskip

\noindent {\bf Response}: I agree that this is quite a natural result. The fact that the Bell operators work only in fixed-$N$ sectors is a consequence of the superselection rules, which here manifest through the conservation of particles. It was not my intention to make the non-fixed $N$ part a highlight, though it is necessary to deal with the SPDC-like problems. The fact that the theory works smoothly for systems with fluctuating number of particles is its nice feature, I trust. I have added the reference to [40] in the section discussing the Bell correlations in the SPDC system.

\bigskip

\bigskip \sep \bigskip

\noindent {\bf Comment II.6}: {\it Before Eq. 2, lambda is a "hidden variable", but Bell's theorem is about local hidden variables. The issue of locality is key here, especially when dealing with indistinguishable particles. How does Bell's theorem formulate in this case? What are the measurement choices in the Bell experiment, since there appears only to be the $\sigma_+$ observable for each particle. That connection should be made explicit, beyond what Fig. 1 hints at.}

\bigskip

\noindent {\bf Response}: Please note that for indistinguishable particles that cannot be addressed individually (for instance by means of some additional external degrees of freedom), only the collective measurements are allowed. Hence the ``traditional'' approach to Bell nonlocality must be rephrased in terms of permutationally invariant operations and then further symmetrized by means of taking the same observable per party. I have significantly expanded the discussion of these issues in Sections 2.1.1 and 2.1.2 to address the concerns risen by the Referee.

\bigskip

\bigskip \sep \bigskip

\noindent {\bf Comment II.7}: {\it $k \in [1,m]$ is perhaps better to say $k \in {1,\ldots,m}$}

\bigskip

\noindent {\bf Response}: I have followed the Referee's remark and made the suggested replacement.

\bigskip \sep \bigskip

\noindent {\bf Comment II.8}: {\it The Cauchy-Schwarz inequality in integral form is used repeatedly in the paper. To make it more self-contained I would suggest to state it in general form the first time it is used.}

\bigskip

\noindent {\bf Response}: Folowiing the Referee's suggestion, I have added both the integral and the discrete form of the Cauchy-Schwarz inequality.

\bigskip \sep \bigskip

\noindent {\bf Comment II.9}: {\it In the 3.1 example, it would be nice if more details about how the ground state for N=100 was found were indicated. }

\bigskip

\noindent {\bf Response}: The ground states were found by the exact diagonalization of the Bosonic Josephson junction Hamiltonian. As suggested by the Referee, a comment on the methodology has been added to the double-well section of the manuscript.

\bigskip \sep \bigskip

\noindent {\bf Comment II.10}: {\it In Figure 1, what would be the physical mechanism by which one would space-like separate the particles in a BEC without changing their (relevant) quantum degrees of freedom?}

\bigskip

\noindent {\bf Response}: I admit that there is no simple way to spatially separate a many-body interacting system without disturbing its quantum degrees of freedom. Although a complete Bell test with spatially separated atoms coming from a BEC is still a distant perspective, there are some techniques that open the way for such future experiments. These are, for instance, the light-sheet method used in Vienna or the single-atom detection with the micro-channel plate used in Canberra or Palaiseau. I have added a proper comment and the references to the discussion presented in Section 2.2.4. \bigskip

\bigskip \sep \bigskip

\noindent {\bf Comment II.11}: {\it References:

R1- In the introduction, two more loophole-free Bell tests were performed in 2015 almost simultaneously with the one cited (and another in 2017) by independent groups.

R2- In the discussion about Eq. 18 the problem about nonlocality depth is highlighted. There exist works that already address this problem in the context of device-independent witnesses of entanglement depth that should be referenced as well.

R3- The triplet of experiments on split BEC https://www.science.org/doi/abs/10.1126/science.aat4590 seems also a good motivation for this paper worthy of mentioning.

R4- The entanglement witness bound $4^{-m}$ in Section 4 needs proper referencing.}

\bigskip

\noindent {\bf Response}: All these suggested references have been added to the manuscript. \bigskip

\bigskip \bigskip \bigskip

\noindent\makebox[\linewidth]{\rule{18cm}{0.4pt}} \noindent

\noindent\makebox[\linewidth]{\rule{18cm}{0.4pt}}

\bigskip

\begin{enumerate}

\item The Section 2 has been completely reworked to address the points risen in the reports of both the Referees. A clear distinction between the establishment of Bell bounds and their testing has been made.

\item The discussion of one particular bound for the A/B nonlocality (Equations 21-23 in the previous version of the manuscript) has been removed, following the Referee's I comment.

\item The problem of spatial separation of atoms forming a BEC prior to the Bell test is now discussed in Section 2.2.4.

\item A remark about the role of GHZ and N00N states in this work has been added to Section 4.

\item The Cauchy-Schwarz inequality is now presented in both the integral and the discrete form.

\item The outline of the methodology of finding the ground state for the double-well case has been added.

\item All the references suggested by Referee II have been added.

\item All other corrections suggested by both Referees have been applied.

\item The figures have been updated by adding the upper index $(q)$ to the correlators.

\end{enumerate}

\end{document}

---

## Round 2 · List of Changes

\begin{enumerate}

\item The Section 2 has been completely reworked to address the points risen in the reports of both the Referees. A clear distinction between the establishment of Bell bounds and their testing has been
made.

\item The discussion of one particular bound for the A/B nonlocality (Equations 21-23 in the previous version of the manuscript) has been removed, following the Referee's I comment.

\item The problem of spatial separation of atoms forming a BEC prior to the Bell test is now discussed in Section 2.2.4.

\item A remark about the role of GHZ and N00N states in this work has been added to Section 4.

\item The Cauchy-Schwarz inequality is now presented in both the integral and the discrete form.

\item The outline of the methodology of finding the ground state for the double-well case has been added.

\item All the references suggested by Referee II have been added.

\item All other corrections suggested by both Referees have been applied.

\item The figures have been updated by adding the upper index $(q)$ to the correlators.

\end{enumerate}

---

## Round 3 · Author Response

Dear Prof. Toth,

Thank you again for your time devoted to handling of my manuscript. I would also like to thank the Referees for their positive comments and their final recommendations.

I fully agree with the revisions suggested by the Referees in this second round. In this resubmission, please find the manuscript with all their remarks taken into account.

I hope that after these modifications, the manuscript could be accepted for the SciPost Physics Core.

With kind regards,
Jan Chwedenczuk

---

## Round 3 · List of Changes

1) In this revised manuscript, the minor corrections have been applied throughout the text. 2) Also, the text has been modified so that it refers to many-body Bell correlations rather than many-body nonlocality.

---

## Editorial Decision

published